# How a student becomes a teacher: learning and forgetting through Spectral methods

**Lorenzo Giambagli**[1,2], **Lorenzo Buffoni**[1], **Lorenzo Chicchi**[1], **Duccio Fanelli**[1]

[1]Department of Physics, University of Florence
[2]Department of Mathematics, University of Namur
{lorenzo.giambagli, lorenzo.buffoni, lorenzo.chicchi,
duccio.fanelli}@unifi.it

## Abstract

In theoretical Machine Learning, the teacher-student paradigm is often employed as an effective metaphor for real-life tuition. A student network is trained on data generated by a fixed teacher network until it matches the instructor's ability to cope with the assigned task. The above scheme proves particularly relevant when the student network is overparameterized (namely, when larger layer sizes are employed) as compared to the underlying teacher network. Under these operating conditions, it is tempting to speculate that the student ability to handle the given task could be eventually stored in a sub-portion of the whole network. This latter should be to some extent reminiscent of the frozen teacher structure, according to suitable metrics, while being approximately invariant across different architectures of the student candidate network. Unfortunately, state-of-the-art conventional learning techniques could not help in identifying the existence of such an invariant subnetwork, due to the inherent degree of non-convexity that characterizes the examined problem. In this work, we take a decisive leap forward by proposing a radically different optimization scheme which builds on a spectral representation of the linear transfer of information between layers. The gradient is hence calculated with respect to both eigenvalues and eigenvectors with negligible increase in terms of computational and complexity load, as compared to standard training algorithms. Working in this framework, we could isolate a stable student substructure, that mirrors the true complexity of the teacher in terms of computing neurons, path distribution and topological attributes. When pruning unimportant nodes of the trained student, as follows a ranking that reflects the optimized eigenvalues, no degradation in the recorded performance is seen above a threshold that corresponds to the effective teacher size. The observed behavior can be pictured as a genuine second-order phase transition that bears universality traits. Code available at: `https://github.com/Jamba15/Spectral-regularization-teacher-student/tree/master`.

## 1 Introduction

Machine learning (ML) technologies nowadays play a pivotal role in systematizing complex data by elucidating how different elements relate to each other. To this end, ML models learn from data by identifying distinctive features, which form their basis for decision-making. This task is accomplished by solving an optimisation problem that seeks to minimise a suitably defined loss function which compares expected and actual output. When operating with deep learning architectures with a feedforward structure, data supplied as input are processed via a repeated sequence of linear and non-linear transformations. The linear transformation, indeed, maps the signal from any given layer

37th Conference on Neural Information Processing Systems (NeurIPS 2023).

to its immediate analogue, following the assigned neural network arrangement. Customarily, the matrix elements of the linear transformation (weights) are the target of the optimization (i.e. loss minimization). In [1] a radically new approach to this learning scheme, which anchors the process to reciprocal space, was first proposed. The training there seeks to modify the eigenvectors and eigenvalues of transfer operators in direct space. By acting on the spectral coordinates no increase in terms of computational and complexity load is produced, as compared to conventional algorithms. Remarkably, the eigenvalues yield a reliable ranking of the nodes, in terms of their contribution to the network performance. Indeed, the absolute value of the eigenvalues proves to be an excellent hallmark of the node's significance to the overall functioning of the trained model. This observation paves the way for an innovative ex-post pruning strategy to cut out unessential neurons and eventually yield smaller models, with almost identical classification abilities and thus more suitable for possible hardware deployment. Starting from these premises, here we will first deepen the understanding of the foundations of spectral learning, by focusing on the key aspects that relate to the underlying optimization protocol. We will prove that the spectral learning method - complemented with an appositely tailored regularization that favors the selection of eigenvalues with reduced magnitude – enables one to isolate an invariant trained subnetwork where the acquired ability to perform the target task is stored using a minimal amount of neurons. This latter is stably retrieved also when starting with an over-parametrized neural network, an outcome that cannot be reached when working with conventional learning algorithms formulated in direct space. To come to this conclusion, we will work under the teacher-student scheme. Concretely, we will prove that the student's ability to mimic a quenched teacher hides in a sub-portion of the whole network which consistently emerges across different architectures of the student candidate network. Remarkably, the student mirrors the teacher in terms of computing neurons, path distribution and topological attributes. As a side remark, we will also generalize the spectral scheme to deal with convoluti<onal models and show that an empirical method for structural pruning, previously proposed in [2], can be rationalized as a simplifying variant of the spectral method above.

## 1.1 Related Works

Neural Network pruning is a relevant topic in Deep Learning [3, 4, 5]. The idea of our method is to a posteriori identify the nodes of the trained network which prove unessential for a proper functioning of the device and cut them out from the ensemble made of active units and related examples can be found in [6]. In this latter work a neuron importance score is calculated as function of the average activation and the related weights. At variance with the method that we propose, the relevance is calculated after training and by using training set related statistical quantities. As we will see no further post-training elaboration are necessary: all the information is stored in the magnitude of the optimized eigenvalues. Another quite effective approach to feature localization in the original space was proposed in [7] utilizing Group Lasso regularization. In [8], following the pioneering approach of [9], the relevance of nodes is assessed by using a Taylor expansion up to order two. It is also worth mentioning that, following the canonical pipeline *Training → Extracting Relevance Indicators → Pruning*, not always yields better performances when compared with model, of the same size of their pruned counterparts, initialized with random weights [10]. The aim of this work is not to conduct a rigorous benchmark against pre-existing pruning methods. Rather, we will prove that the spectral attributes of the neural network enable one to extract the relevant computational core for the task under inspection.

## 2 Spectral Parametrization of Feedforward Fully-connected layer

To establish a common understanding of the formalism, let us start by describing the linear action of a fully connected layer. Introduce $x_j^{(k-1)}$, with $j \in 1 \ldots N_{k-1}$ and $k \in 1 \ldots \ell$, to represent the activity at the $j$-th neuron of layer $k-1$. The linear action of the layer is expressed as a vector-matrix multiplication: $z_i^{(k)} = \sum_j w_{ij}^{(k)} x_j^{(k-1)}$ where the matrix components $w_{ij}^{(k)}$ set the weight of the connections from node $j$ to $i$. In vector notation yields: $\mathbf{z}^{(k)} = \mathbf{w}^{(k)} \cdot \mathbf{x}^{(k-1)}$. We then apply a chosen element-wise non-linearity to the vector $\mathbf{z}^{(k)}$ to obtain $\mathbf{x}^{(k)}$ and repeat this operation for all the layers $k \in 1 \ldots \ell$ to build the neural network architecture of our choice.
It is a well known interpretation [11] that this vector-matrix multiplication results in the projection of $\mathbf{x}^{(k-1)}$ along each feature $\mathbf{w}_i^{(k)}$, which corresponds to the linear activation of the corresponding

neuron (ignoring the bias addition). In this framework, one can speculate that the relevance of each feature can be assessed through a further set of scalar parameters, associated to the destination node, that modify the underlying optimization process. These latter parameters, denoted $\lambda_i^{(k)}$ for reasons that will become clear in the following section, get initialized to one and leverage the projection along each feature $\mathbf{w}_i^{(k)}$. The rescaled linear transfer of information then becomes:

$$z_i^{(k)} = \lambda_i^{(k)} \sum_j w_{ij}^{(k)} x_j^{(k-1)} \tag{1}$$

or equivalently, in vector notation:

$$\mathbf{z}^{(k)} = \lambda^{(k)} \odot (\mathbf{w}^{(k)} \cdot \mathbf{x}^{(k-1)}) \tag{2}$$

where $\odot$ stands for the Hadamard (or element-wise) product. As we shall prove in the following, this seemingly simple reparametrization holds unexpected capabilities and profound connections with graph theory.

To this end, we begin by noting that layer $k$ can be viewed as a bipartite directed graph connecting the nodes in layer $k-1$ to those in layer $k$. This graph can be fully described in terms of an *adjacency matrix* $A^{(k)}$, which encodes all the relevant information on existing connections. $A^{(k)}$ is a $(N_{k-1} + N_k) \times (N_{k-1} + N_k)$ matrix. If we label the neurons belonging to layer $k-1$ from 1 to $N_{k-1}$ and those in layer $k$ from $N_k + 1$ to $N_{k-1} + N_k$, the elements $A_{lm}^{(k)}$ exemplify the weighted connection from node $m$ to node $l$, with $l, m \in 1 \ldots N_{k-1} + N_k$. Due to the directed nature of the graph, the structure of $A^{(k)}$ is lower-block diagonal and contains the values $w_{ij}^{(k)}$ of the network layer in its nonzero terms. For a more detailed insight into the actual matrix structure, we refer to the extended discussion in the Supplementary Material. In this framework, the activity of nodes across the graph can be described through a vector, termed $v$ for clarity, made of $N_{k-1} + N_k$ components, where $v_m$ points to the activity of node $m$. This will refer to a neuron in layer $k-1$ if $m \leq N_{k-1}$, or, conversely, to layer $k$ if $N_{k-1} + 1 \leq m \leq N_k + N_{k-1}$. The structure of such vector for activities localized on layer $k-1$, and before the linear transfer gets applied, is shown in Eq.(3).

The feedforward transfer from layer $k-1$ to layer $k$ can be hence described as the action of the square matrix $A^{(k)}$ on the vector $v$. The resulting vector components, which are connected to the transferred activity $\mathbf{z}^{(k)}$, will be equal to 0 for $m \leq N_{k-1}$. This can be expressed mathematically as:

$$v = \begin{pmatrix} x_1^{(k-1)} \\ \vdots \\ x_{N_{k-1}}^{(k-1)} \\ 0 \\ \vdots \\ 0 \end{pmatrix} \quad \Rightarrow \quad \begin{pmatrix} 0 \\ \vdots \\ 0 \\ z_1^{(k-1)} \\ \vdots \\ z_{N_k}^{(k-1)} \end{pmatrix} = A^{(k)} v = A^{(k)} \begin{pmatrix} x_1^{(k-1)} \\ \vdots \\ x_{N_{k-1}}^{(k-1)} \\ 0 \\ \vdots \\ 0 \end{pmatrix} \tag{3}$$

The above graph-oriented way of describing the activity paves the way to a different parametrization of the linear transfer between layers, of which Eq.(1) represents a special case. The spectral parametrization builds on the observation that another class of adjacency matrices can be written, whose action on vector $v$ is equivalent to that illustrated above. This latter matrix $\tilde{A}$ can be assembled starting from two other matrices: a diagonal *eigenvalues* matrix with entries $\lambda_j^{(k)\ in}$ for the first $N_{k-1}$ elements and $\lambda_i^{(k)\ out}$ for the last $N_k$ and one lower-block triangular *eigenvectors* matrix with nonzero elements $\phi_{ij}^{(k)}$ and unitary diagonal.

This reparametrization is made possible by the fact that the feedforward action is solely encoded in the lower-block diagonal elements, and will always operate on vectors whose structure is depicted in Eq.(3). The block structure of the matrix $\Phi^{(k)}$ is the one responsible for the feedforward behaviour of the resulting adjacency matrix. Specifically, this structure enables activity localized in the $k-1$ layer (i.e., vectors whose first $N_{k-1}$ components are non-zero) to be transferred to the $k$-th layer, resulting in a vector whose last $N_k$ components are non-zero. Additionally, it is worth mentioning that the inverse of $\Phi^{(k)}$ can be computed analytically and it is equal to $(\Phi^{(k)})^{-1} = 2\mathbb{I}_{N_{k-1}+N_k} - \Phi^{(k)}$.

Using this property, we can explicitly express the result of the spectral composition $\tilde{A}^{(k)} =$

$\Phi^{(k)}\Lambda^{(k)}(\Phi^{(k)})^{-1}$ as a lower-triangular matrix, whose element realizing the linear transfer of information as in Eq.(3) can be called $\tilde{w}_{ij}$ (see Supplementary Material for more details). Previous research [1, 12, 13] has demonstrated that we can write the elements $\tilde{w}_{ij}$ in terms of the off-diagonal elements of the eigenvector matrix $\Phi^{(k)}$ and the diagonal eigenvalue matrix $\Lambda^{(k)}$, specifically:

$$\tilde{w}_{ij}^{(k)} = (\lambda_j^{(k)in} - \lambda_i^{(k)out})\phi_{ij}^{(k)} \tag{4}$$

The matrix $\tilde{A}$ acts as an equivalent feedforward network, transferring $k-1$-layer localized activity to the following layer through a simple vector-matrix multiplication. By substituting $\tilde{A}$ for $A$ in Eq.(3), we can express the linear activity of neurons in the $k$-th layer, $z_i^{(k)}$, in terms of the spectral parameters introduced earlier and the $k-1$ layer activity $x_j^{(k-1)}$, namely:

$$z_i^{(k)} = \sum_{j=1}^{N_{k-1}} (\lambda_j^{(k)in} - \lambda_i^{(k)out})\phi_{ij}^{(k)}x_j^{(k-1)} \quad \forall i \in 1 \dots N_k \tag{5}$$

or, in vector notation

$$\mathbf{z}^{(k)} = \boldsymbol{\phi}^{(k)} \cdot (\boldsymbol{\lambda}^{(k)in} \odot \mathbf{x}^{(k-1)}) - \boldsymbol{\lambda}^{(k)out} \odot (\boldsymbol{\phi}^{(k)} \cdot \mathbf{x}^{(k-1)}) \quad \forall i \in 1 \dots N_k \tag{6}$$

This simple decomposition allows for an intuitive understanding of the role of eigenvalues and eigenvectors, keeping the computational cost almost invariant, since the inverse of the eigenvector matrix $\Phi^{(k)}$ is given analytically. It is now immediate to see that, by setting $\lambda_j^{(k)in} = 0$ for all $j \in 1 \dots N_{k-1}$, we recover Eq.(1). The features can now be seen as components of the eigenvectors of the adjacency matrix describing the sub-network made by layers $k$ and $k-1$. Their relative weight, instead, can be interpreted as the magnitude of the eigenvalue $\lambda_j^{(k)out}$. Since the feedforward fully-connected nature of the transfer from layer $k-1$ to $k$ is encoded in the structure of $\Phi^{(k)}$, the entries of this latter matrix enable one to resolve the relationship between different eigenvector structures and dive into the obtained network topology. In Supplementary Material, we will show how a Convolutional Neural Network (CNN) scheme can be conceptualized as a genuine eigenvalue training for a peculiar structure of $\Phi^{(k)}$. Here the eigenvalues gauge the relative importance of the employed convolutional filters.

As already shown in the literature (see [14] ), computing the gradient with respect to the eigenvalues and eigenvectors yields a node-related set of scalars (the magnitude of the optimized eigenvalues) that can be exploited for structural pruning. In this work, we extend this idea by introducing a spectral regularization term in the loss function. Using the simplified spectral formulation of Eq.(2), the gradient will be computed with respect to the parameters $\{\boldsymbol{\lambda}^{(k)}, \boldsymbol{\phi}^{(k)}\}_{k \in 1 \dots \ell}$ and feature localization forced by adding a $L_2$ penalty in the loss function - in light of the above interpretation that identifies in $\lambda$ a feature weight. We will show that this learning strategy yields a non-trivial effect in that it enables to consistently identify an invariant information bulk stored inside the trained network. The invariance is assessed with respect to the initial complexity of the network employed. The characteristic of this minimal computing core will be explored working in a controlled teacher-student framework where the complexity of the teacher, namely the target function, is set to be lower than the student, the optimized network.

Remarkably several topological properties of the teacher can be inferred via post-training student analysis if spectral parametrization and regularization are employed. This is at variance with the conventional training scheme, where a $L_2$ regularization that aims at inter-layers connections produces sparsity effects but yields no invariant core nor topological similarities with the teacher.

## 3  Definition of the experimental framework

The teacher-student framework is a widely recognized concept in the field of theoretical deep learning, and has gained significant attention due to its effectiveness in knowledge distillation [15, 16, 17], and theoretical insight [18, 19, 20]. Introduced by Hinton et al.[21],it employs a teacher model to guide the learning of a student model. By using a softened probability distribution from the teacher model as labels, the student model is encouraged to imitate the teacher's behaviour.

In this work, we will use this particular setting and focus on three different learning regimes. These latter can be hierarchically ranked depending on whether the complexity of the function that the

student network learns is greater than, less than, or equal to the complexity of the teacher network function that the student is attempting to regress.

The teacher network is characterized by specific topological features that reflect its inherent structure and path distribution. As a follow-up of the analysis, we will compare these characteristics to those recovered by the student network operated under the spectral paradigm. The degree to which they are preserved in the latter will be one of the metrics that we shall use to determine the effectiveness of the learning process. To ensure simplicity and generality in our analysis, we will adopt a teacher network with two hidden layers and a scalar output. The weights that refer to the final layer are all set to one. The proposed formulation is hence inspired by the theory of soft committee machines [22, 23], but with an additional layer to increase the problem's complexity.

The dataset was generated by choosing the probability distribution of the Instance space $p(x)$ to be a Standardized Gaussian in $\mathbb{R}^{10}$, namely $p(x) = \mathcal{N}(x; \mu = 0, \Sigma = \mathbb{I})$. In this framework, the teacher network $T(x)$ is used as a Supervisor defining a conditioned probability distribution $g(y|x) = \delta(y - T(x))$. The dataset is thus composed by the collection to tuples $\mathcal{D} = \{(x^{(i)}, y^{(i)})_{i \in 1 \ldots |D|} \mid (x^{(i)}, y^{(i)}) \sim p_{data}(x, y) = p(x)g(y|x)\}$ and has a number of elements $|D| = 1.3 \, 10^4$.

The dimensions of the teacher layers are respectively set to $10 - 20 - 20 - 1$ and are initialized according to a standard Glorot Uniform distribution [24]. The student network $S(x)$, on the other hand, is formed by a two hidden layers deep neural network with dimension $10 - h - 20 - 1$ where $h$ is a variable integer that will be tuned across experiments. The second hidden layer is taken to be a standard fully connected layer (the gradient will be computed with respect to the connections). The first hidden layer, whose dimension $h$ can be changed at will, can be either parametrized as a fully-connected layer or as its spectral analogue (the eigenvalues $\boldsymbol{\lambda}^{(1)}$ and eigenvectors components $\boldsymbol{\phi}^{(1)}$ are thus the target of the optimization), depending on the specific aims of the analysis. Accordingly, the loss function will be different depending on the layer employed. Namely, we will use

$$L = \frac{1}{|D|} \sum_{i \in 1}^{|D|} (T(x^{(i)}) - S(x^{(i)}))^2 + \alpha_w \sum_{k \in 1,2} L_2(\boldsymbol{w}^{(k)}) \tag{7}$$

for the student parametrized in the standard way, and

$$L = \frac{1}{|D|} \sum_{i \in 1}^{|D|} (T(x^{(i)}) - S_\lambda(x^{(i)}))^2 + \alpha_\lambda L_2(\boldsymbol{\lambda}^{(1)}) + \alpha_\phi L_2(\boldsymbol{\phi}^{(1)}) + \alpha_w L_2(\boldsymbol{w}^{(2)}) \tag{8}$$

for the student which bears a spectral parametrization of the first hidden layer, here denoted by $S_\lambda$ for clarity. Note the $L_2$ penalty term that, as discussed above, acts as a regularization.

To focus on the impact of the spectral parametrization of the layer, we will initialize $S$ and $S_\lambda$ such that the set of links is identical but still randomized. We accomplish this by setting $\lambda_i^{(1)} = 1$ for $i \in 1 \ldots h$ and $\phi_{ij}^{(1)} = -w_{ij}^{(1)}$ for all $i \in 1 \ldots h$ and $j \in 1 \ldots 10$. The $w_{ij}^{(1)}$ are sampled from a Glorot distribution, ensuring that there is no bias in the optimization due to differences in the initialization of the inter-nodes connections. For each run, we will use the minibatch stochastic gradient descent algorithm Adam, with a batch size of 300 and 500 for $S_\lambda$ and $S$, respectively, and a total of 2000 epochs. The relevance of the features extracted by $S$ and $S_\lambda$ will be computed as the norm of the vectors that modulate the information to be conveyed at the destination nodes on the layer of variable size $h$. More specifically, we can extract from the conventionally parametrized student $S$, the following indicator bound to node $i$:

$$\mathcal{W}_i = \left( \sum_{j=1}^{h} w_{ij}^2 \right)^{1/2} \tag{9}$$

For the network $S_\lambda$, on the other hand, we will employ the following expression:

$$\mathcal{L}_i = \lambda_i \left( \sum_{j=1}^{h} \phi_{ij}^2 \right)^{1/2} \tag{10}$$

which scales proportionally to the eigenvalue entry $\lambda_i$. It is clear that these quantities hold for every $i \in 1 \ldots h$, and both support the idea that the larger the $L_2$ magnitude of the connections, the

more relevant the associated processing node. Notice, however, that this latter quantity needs to be appropriately rescaled for the corresponding eigenvalue's magnitude for fair comparison when operating under spectral parametrization. We speculate that by imposing a node-wise localization of the features when working under the usual parametrization results in a cumbersome excercise which cannot be easily performed. On the other hand, this goal can be instead accomplished when working under the spectral parametrization. We also speculate that the same effect could be induced on the input features when using all the spectral components, i.e. when the $\lambda^{in}$ in Eq.(4) are not left untrained.

## Formulation of the algorithm

The results presented in this paper are obtained by using an algorithm that can be summarized as follows:

- Replace the conventional feedforward, fully-connected layers with the so called *Spectral* layers, where the links are parameterized as in Equation (4).

- Initialize $\lambda^{(k)in}$ to 0 (and leave it untrained), set $\lambda^{(k)out}$ to 1, and initialize $\phi^{(k)}$ by using the Glorot distribution [24].

- Apply $L_2$ regularization to $\lambda^{(k)out}$ and $\phi^{(k)}$, and proceed with the optimization.

- Examine the entries $\mathcal{L}_i$ for each node $i$ from Equation (10); a larger value indicates higher node relevance.

The implementation of the *Spectral* layer and the Structural pruning functions can be found in `https://github.com/Jamba15/SpectralTools` and Supplementary Material.

## Results

In order to assess the effectiveness of the introduced parametrization and regularization, the dimension of the hidden layer $h$ has been varied within the interval $\mathcal{I} = $ `[10, 20, 40, 60, 100, 200, 500, 700, 1000]`. For each $h \in \mathcal{I}$, a total of 30 trials have been conducted, and train and test losses have been recorded for statistical purposes. This evaluation has been performed for both $S$ and $S_\lambda$, which we will now denote as $S(h)$ and $S_\lambda(h)$ respectively, with $h \in \mathcal{I}$. The teacher function $T(x)$ is fixed as described above and the student gets trained over 2000 epochs, to ensure a consistent and proper convergence. The average test loss, evaluated on a different set of 1000 samples distributed according to the same $p_{data}(x, y)$, is computed. For all $h > 20$, the average mean squared error (MSE) of $S(h)$ is $\langle \text{MSE}(S(h)) \rangle = 9 \pm 1 \times 10^{-3}$ (with a one-sigma error), and the average MSE of the predictions of $S_\lambda(h)$ reads $\langle \text{MSE}(S_\lambda(h)) \rangle = 9 \pm 3 \times 10^{-3}$. Note that averages here are taken over the 30 repetitions of the experiment. It is worth emphasizing that these values stay stable across all choices of $h$ above 20. This implies that both models are operated in a properly converging regime, where the two different parametrizations prove equivalent in terms of test error. Building upon this, we can now examine the properties of the network after training in light of the two employed parametrizations. To assess feature localization within the network, we generated aggregated histograms of the scalars (Eq. 9) and (Eq. 10). This aggregation was performed across all trials for a fixed hidden layer size, $h$. The range scanned by the aforementioned scalar entries was rescaled in such a way that both $\mathcal{W}$ and $\mathcal{L}$ lie in the same range $[0, 1]$. This enables for a meaningful comparison between the two distributions. Figure 1 demonstrates a stark difference between the empirical distributions of the two computed scalars. The blue histograms stands for (Eq. 10), which refer to the spectral attribute $\mathcal{L}$, while the orange histograms represent the feature norm for a conventional regularized training scheme. The latter exhibits a prototypical behavior, with the distribution slowly shifting towards zero as the size of the hidden layer increases. On the other hand, the former displays a more marked dependence on the imposed size $h$ of the variable hidden layer. In the overparameterized ($h > 20$) student regime, we observe the emergence of a peak in the first bin, which corresponds to values of the feature parameter close to zero. This observation qualifies as a significant change as compared to the underparameterized regime, where the distribution only contains non-zero values of $\mathcal{L}_i$. On the other hand, the behaviour of $\mathcal{W}$ remains qualitatively similar across different sizes of system being trained. The significance of the zero-peaked $\mathcal{L}$ values

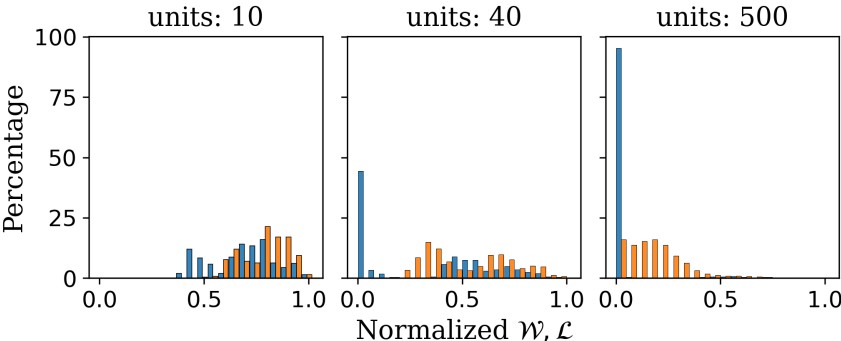

Figure 1: Histogram for the quantities (Eq. 10) (in blue) and (Eq. 9) (in orange) for the first hidden layer of the student. The scalars entries are divided by the maximum of the sample so that the two distributions lies in the interval $[0, 1]$.

is straightforward: these are the features (i.e. the neurons) that can be safely discarded without affecting the generalization properties of the network $S_\lambda(h)$. To support this interpretation, we plot the average dimension size of the first hidden layer, considering only the non-zero features, in the left panel of Figure 2. Strikingly enough, we observe the emergence of an invariant processing core within the student, the dimension of which is closely connected to that of the teacher (in this case, $h = 20$). Despite achieving the same test accuracy, the two networks ($S$ and $S_\lambda$) exhibit distinct topologies and levels of interpretability. The former demonstrates a sparse structure that distributes information across the entire set of connections, while the latter, obtained as a byproduct of the spectral training complemented with a suitable regularization, drives the information processing towards a minimal and identifiable subnetwork core. The additional feature weighting parameter, which can be interpreted as a particular case of a Spectral parametrization of the transfer operator, could be effectively utilized, along with regularization, to identify the Winning Lottery Ticket features within the network. We speculate that an iterative approach, similar to the one described by Frankle and Carbin [25] but focused on nodes rather than links, may be feasible, but we leave this deepening for future investigation.

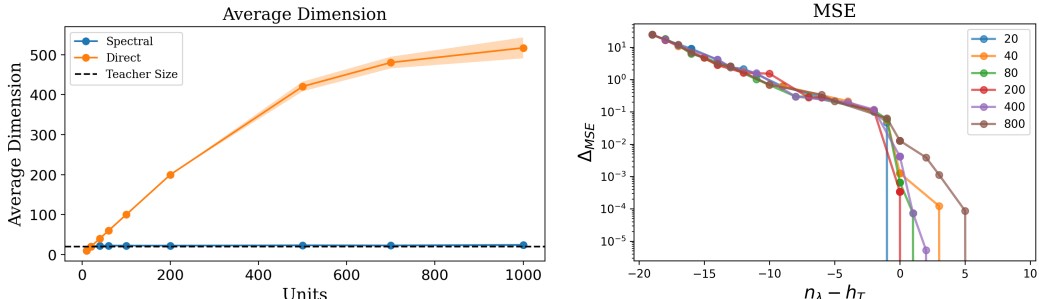

Figure 2: (Left) Average dimension of the first hidden layer after the nodes associated to zero values of $\mathcal{W}$ or $\mathcal{L}$ have been removed. The average is taken across the 30 trials and the shaded region refer to one standard deviation. (Right) The deviation of the mean squared error (MSE) is plotted against $n_\lambda - n_T$. Here $n_\lambda$ stands for the number of neurons that are left after pruning, while $n_T$ refer to the actual size of the teacher. A phase transition-like behaviour occurs when the size of the first hidden layer of the student matches that of the teacher network. The vertical scale is logarithmic.

To ensure that the achieved conclusions are not influenced by the size of the second hidden layer, various student configurations have been tested, yielding equivalent results. Specifically, the size of the second layer has been modulated within a wide range. The size of the preceding layer and that of the corresponding layer in the teacher network are included within this range. In every case, the size of the retrieved computational core remains constant, across different initial choices of the variable hidden layer size. The residual size reflects the inherent complexity of the teacher. As such, it could be thus different from the size of the teacher's homologous layer.

To provide a complementary view, we also estimate the deviation of the mean squared error (MSE) computed after pruning (i.e. upon removing unessential nodes as ranked via $\mathcal{L}$) from the corresponding value recovered at the end of training, for the full network. This latter quantity is plotted in the right panel of Figure 2 against $n_\lambda - n_T$ where $n_\lambda$ represents the number of residual neurons and $n_T$ refers to the number of nodes in the teacher (here $h_T = 20$). The invariant core exhibits remarkable robustness when neurons corresponding to low $\mathcal{L}$ values are removed. However, as soon as the minimal bulk is perturbed, the MSE deviation starts growing regardless of the initial size of the layer, thereby revealing a universal scaling profile with respect to $h$. More specifically, the deviation follows an approximately exponential decrease, and the curve $\Delta_{MSE}$ vs. $n_\lambda - h_T$ exhibits a critical point at $n_\lambda - h_T = 0$, where a sudden change in the generalization performance of $S_\lambda$ is observed. Our results have been confirmed with reference to other more realistic datasets. The same conclusions have been reached, both in terms of the emegerence of the invariant structure and the phase-transition like behaviour. In particular we tested our framework by using (i) Shuffled Fashion MNIST (ii) Shuffled MNIST (iii) two hidden layer teacher but with heterogeneous hidden size (20-40-20-1) (iv) California Housing (v) CIFAR-100 with a pretrained ResNet50 backbone enhanced with three Spectral Layers and Batch Normalization. The corresponding results can be found in Figure 3

It is important to emphasize that the structure of the conventionally parametrized network is fundamentally different, and due to its sparsity, it is impossible to conduct a similar analysis for extracting a compact invariant subnetwork. After applying the node removal strategy to $S_\lambda$, we can further analyze the relationship between the pruned student network and the teacher network. Specifically, we investigate whether any properties of the teacher, other than its layer size, can be inferred from the analysis of the invariant core within the student network $S_\lambda$. We focus on a natural property given the feedforward structure of both networks: the path magnitude. To examine the magnitude of each path in the compacted network from a given input node to a specific neuron in the second hidden layer, we construct the expression:

$$\tilde{w}_{i_0,i_1}^{(1)} \cdot \tilde{w}_{i_1,i_2}^{(2)} = \Gamma_{i_0,i_2}^{i_1}, \quad \forall i_0 \in 1\ldots N_0 = 10, i_1 \in 1\ldots h, i_2 \in 1\ldots N_2 = 20 \qquad (11)$$

Here, $\tilde{w}_{i_0,i_1}^{(1)}$ represents the weight between input node $i_0$ and hidden layer neuron $i_1$, and $\tilde{w}_{i_1,i_2}^{(2)}$ represents the weight between hidden layer neuron $i_1$ and second hidden layer neuron $i_2$. $\Gamma_{i_0,i_2}^{i_1}$ then corresponds to the path from $i_0$ to $i_2$ passing through neuron $i_1$. To account for the large amount of permutation invariance in the problem, we transform the tensor $\Gamma_{i_0,i_2}^{i_1}$ into a vector $\vec{\Gamma}$ of size $N_{tot} = N_0 \cdot N_1 + N_1 \cdot N_2$, where the components $\Gamma_i$ range from 0 to $N_{tot}$.

Next, we sort the components of the vector in ascending order and plot them for the teacher network in orange, and for the student network in blue and green (Figure 4). The values are plotted with respect to fraction of the vector components obtained by mapping the index of the sorted vector between zero and one, enabling a meaningful comparison between the networks (due to different number of nodes).

Upon visual inspection of panel (a) in Figure 4, it is evident that the combined effect of spectral parametrization and regularization yields the emergence of an invariant core within the network. This core can be easily retrieved and exhibits a path structure that closely aligns with that of the teacher network.

In contrast, when analyzing the sparse structure of the conventionally trained and $L_2$ regularized network, a significantly different path structure is seen, as depicted in panel (b) of Figure 4. Specifically, the edges of the path magnitudes differ, as does the overall distribution.

## 4    Limitations and future work

While our study provides valuable insights into the regularization and pruning of neural networks, there are some limitations that must be considered. First, the experiments so far conducted employ a straightforward student network architecture. Although this choice allows for a cleaner interpretation of the underlying principles, one should repeat the analysis assuming more complex architectures to draw general conclusions. We are in the process of extending our parametrization techniques to more advanced structures so as to address this limitation. Second, our regularization and pruning scheme have only been validated on relatively simple datasets. While this demonstrates the effectiveness of our approach within a controlled setting, it leaves open the question of how well the method

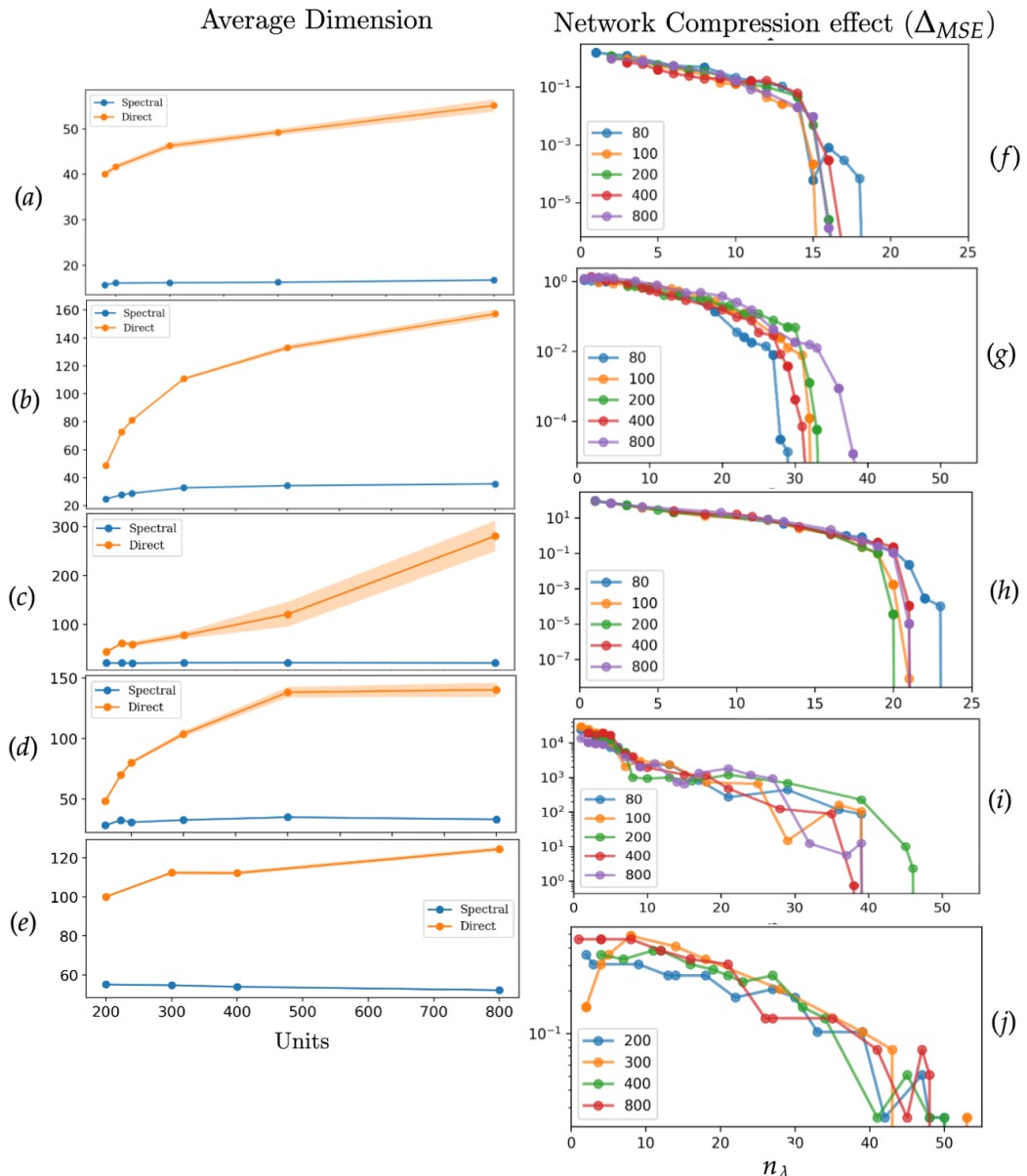

Figure 3: Average dimension of the hidden layer (a-e) and effect on the $\Delta_{MSE}$ (f-j) using the node removal strategy mentioned in the paper. The dataset presented are: Shuffled Fashion MNIST (panels $a$ and $f$), Shuffled MNIST ($b$ and $g$), two hidden layer teacher but with heterogeneous hidden size (20-40-20-1) (panels $c$ and $h$), and California Housing (on panels $d$ and $i$) and CIFAR-100 with a ResNet50 pretrained backbone (panels $e$ and $j$). In this case no reference dimension is shown. Regarding the Average dimension of the hidden layer the accuracy and loss of the Spectral and Classical students are the same for each size (within the statistical error, namely the one standard deviation variance within the 30 trials done for each 'Units' value).

generalizes to more complex, real-world datasets. Future work will involve testing the proposed techniques on larger and more reach datasets once we move to more advanced architectures. Third, in the current work, we solely focused on ranking the nodes belonging to the hidden layers with a combined optimization of eigenvalues and eigenvectors. An extension of this approach amounts to also including in the analysis neurons associated to the input space. This latter could provide valuable insights on key relevant features, particularly when dealing with high-dimensional or complex data.

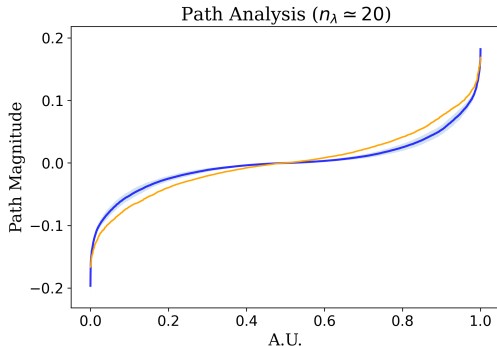 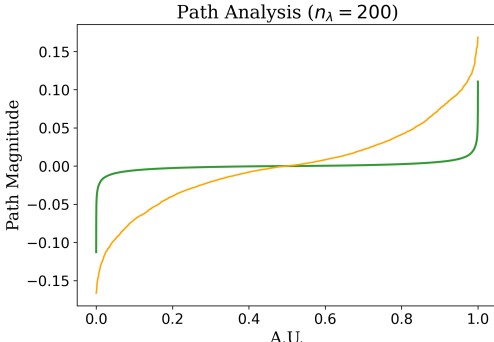

(a) Path comparison between compressed (Pruned) Spectral student (blue) and the teacher (orange).

(b) Path comparison between the full Direct space trained (green) and the teacher (orange).

Figure 4: Comparison of the path magnitude between the input and second hidden layer of the teacher, displayed in orange, and those obtained for every other case study here considered. The first hidden layer in the student network is set to 200 but equivalent results hold for every other $h \in \mathcal{I}$. In panel (a) the first hidden Spectral layer is compressed by removing the nodes that populate with the peak of the normalized distribution of $\mathcal{L}$ reaching $n_\lambda = 20$ neurons. In panel (b) no compression is done. As can be seen the correct distribution cannot be retrieved via the usual Direct approach.

## 5    Conclusions

In summary, our study illustrates the advantages of a novel approach to Deep Neural Networks (DNNs) training which is anchored on the spectral domain. The eigenvector and eigenvalues of the underlying linear transfer operators are the target of the optimization. The proposed method weights feature relevance in each layer through a dedicated indicator which scales as the eigenvalues. When regularized, this latter quantity enables us to identify a minimal, efficient subnetwork within the original structure that retains full network performance upon pruning. Tested using the teacher-student paradigm, we found that this reduced network recovers the same topological features as the original teacher network. This approach outperforms conventional weight decay and classical parametrization, opening up new possibilities for the optimization and scalability of DNNs especially when working in the overparametrized regime.

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

## Supplementary Material - How a Student becomes a Teacher: learning and forgetting through Spectral methods

## A  Spectral Reparametrization

We will give here some more details about the spectral parametrization, with specific regards to the construction of the underlying adjacency matrices. To this end, we begin by noting that layer $k$ of a fully connected neural network, as noted in the main text, can be viewed as a bipartite directed graph connecting the nodes in layer $k-1$ to those in layer $k$. This graph can be fully described in terms of an *adjacency matrix* $A^{(k)}$, which encodes all the relevant information on existing connections. $A^{(k)}$ is a $N_{k-1} + N_k \times N_{k-1} + N_k$ matrix. If we label the neurons belonging to layer $k-1$ from 1 to $N_{k-1}$ and those in layer $k$ from $N_k + 1$ to $N_{k-1} + N_k$, the elements $A_{lm}^{(k)}$ exemplify the weighted connection from node $m$ to node $l$, with $l, m \in 1 \dots N_{k-1} + N_k$.

Due to the directed nature of the graph, the structure of $A^{(k)}$ is lower-block diagonal and can be represented as:

$$A^{(k)} = \begin{pmatrix} 0 & & \cdots & & & 0 \\ \vdots & & & & & \\ 0 & & \cdots & & 0 & 0 \\ w_{11}^{(k)} & \cdots & w_{1N_{k-1}}^{(k)} & \vdots & & \vdots \\ \vdots & & \vdots & & & \\ w_{N_k 1}^{(k)} & \cdots & w_{N_k N_{k-1}}^{(k)} & 0 & \cdots & 0 \end{pmatrix} \tag{A.1}$$

In this framework, the activity of nodes across the graph can be described through a vector, termed $v$ for clarity, made of $N_{k-1} + N_k$ components, where $v_m$ points to the activity of node $m$. This will refer to a neuron in layer $k-1$ if $m \le N_{k-1}$, or, conversely, to layer $k$ if $N_{k-1} + 1 \le m \le N_k + N_{k-1}$. The structure of such vector for activities localized on layer $k-1$, and before the linear transfer gets applied, is shown in Eq.(A.2).

The feedforward transfer from layer $k-1$ to layer $k$ can be hence described as the action of the square matrix $A^{(k)}$ on the vector $v$. The resulting vector components, which are connected to the transferred activity $\mathbf{z}^{(k)}$, will be identically equal to zero for $m \le N_{k-1}$. This can be expressed mathematically as:

$$v = \begin{pmatrix} x_1^{(k-1)} \\ \vdots \\ x_{N_{k-1}}^{(k-1)} \\ 0 \\ \vdots \\ 0 \end{pmatrix} \quad \Rightarrow \quad \begin{pmatrix} 0 \\ \vdots \\ 0 \\ z_1^{(k-1)} \\ \vdots \\ z_{N_k}^{(k-1)} \end{pmatrix} = A^{(k)} v = A^{(k)} \begin{pmatrix} x_1^{(k-1)} \\ \vdots \\ x_{N_{k-1}}^{(k-1)} \\ 0 \\ \vdots \\ 0 \end{pmatrix} \tag{A.2}$$

As explained in the main text, the above graph-oriented way of describing the activity paves the way to a different parametrization of the linear transfer between layers.

The spectral parametrization builds on the observation that another class of adjacency matrices can be written, whose action on vector $v$ is equivalent to that illustrated above. This latter matrix can be assembled starting from two other matrices: a diagonal *eigenvalue* matrix and one lower-block triangular *eigenvector* matrix.

This reparametrization is made possible by the fact that the feedforward action is solely encoded in the lower-block diagonal elements, and will always operate on vectors whose structure is depicted in

Eq.(A.2). Take then two matrices, denoted respectively $\Phi^{(k)}$ and $\Lambda^{(k)}$, with the following structure:

$$\Phi^{(k)} = \begin{pmatrix} 1 & & \cdots & & & & 0 \\ \vdots & \ddots & & & & & \\ 0 & & 1 & 0 & & & 0 \\ \phi_{11}^{(k)} & \cdots & \phi_{1N_{k-1}}^{(k)} & 1 & & & \vdots \\ \vdots & & \vdots & 0 & \ddots & & 0 \\ \phi_{N_k 1}^{(k)} & \cdots & \phi_{N_k N_{k-1}}^{(k)} & 0 & \cdots & & 1 \end{pmatrix} \quad \Lambda^{(k)} = \begin{pmatrix} \lambda_1^{(k)\ in} & 0 & & \cdots & & & 0 \\ 0 & \ddots & & & & & \\ & & \lambda_{N_{k-1}}^{(k)\ in} & 0 & & & 0 \\ \vdots & & & \lambda_1^{(k)\ out} & & & \vdots \\ & & & 0 & \ddots & & 0 \\ 0 & & \cdots & & 0 & \cdots & \lambda_{N_k}^{(k)\ out} \end{pmatrix}$$

$$(A.3)$$

the block structure of matrix $\Phi^{(k)}$ is the one responsible for the feedforward arrangement of the resulting adjacency matrix. Specifically, this structure enables activity localized in the $k-1$ layer (i.e., vectors whose first $N_{k-1}$ components are non-zero) to be transferred to the $k$-th layer, resulting in a vector whose last $N_k$ components are non-zero. Additionally, it is worth mentioning that the inverse of $\Phi^{(k)}$ can be computed analytically and it is equal to $(\Phi^{(k)})^{-1} = 2\mathbb{I}_{N_{k-1} \times N_k} - \Phi^{(k)}$.

Using this property, we can explicitly express the result of the spectral composition $\tilde{A}^{(k)} = \Phi^{(k)}\Lambda^{(k)}(\Phi^{(k)})^{-1}$ as a lower-triangular matrix, shown in Eq.(A.4).

$$\tilde{A}^{(k)} = \begin{pmatrix} \lambda_1^{(k)\ in} & & \cdots & & & & 0 \\ \vdots & \ddots & & & & & \\ 0 & & \lambda_{N_{k-1}}^{(k)\ in} & 0 & & & 0 \\ \tilde{w}_{11}^{(k)} & \cdots & \tilde{w}_{1N_{k-1}}^{(k)} & \lambda_1^{(k)\ out} & & & \vdots \\ \vdots & & \vdots & 0 & \ddots & & \\ \tilde{w}_{N_k 1}^{(k)} & \cdots & \tilde{w}_{N_k N_{k-1}}^{(k)} & 0 & \cdots & & \lambda_{N_k}^{(k)\ out} \end{pmatrix} \qquad (A.4)$$

We can the conclude that the action of this matrix on vector $v$ is analogous to that exemplified in Eq.(A.1) provided the weights $\tilde{w}_{ij}^{(k)}$ are parametrized according to the spectral recipe as introduced in the main text.

## B    Generalization to Convolutions

In the following, we will show how the spectral parametrization can be extended to a Convolutional Neural Network (CNN) Layer. In order to do that we shall first describe the CNN action as a classical vector matrix multiplication. Working with CNNs, once the size of the filter is fixed, one needs to specify (i) the $x, y$ striding ($s_{x,y}$), i.e. how much each filter is shifted in each direction during the convolution, and (ii) the $x, y$ padding ($p_{x,y}$) i.e. how many zeros are added at each side of the input so that the convolution starts and ends with a given "offset". More specifically focus on the notation introduced Figure B.1. We are aware of the fact that we are showing a simplified case of the general convolutions, where also different channels are present. However the underling math is basically the same and, for a matter of clarity, only this simplified yet general case is presented. The same operation can be framed in terms of a matrix-vector multiplication as soon as every parameter is fixed. To this end fix the padding to 0 and the striding to 1, without loss of generality. The convolution of the input $x^0$ with a filter $w$ can be written as the action of a matrix $\mathcal{M}(w)$ on the input written with an equivalent vector notation. $\mathcal{M}(w)$ is a matrix with a peculiar structure called *Toeplitz matrix*. A paradigmatic example is shown for the aforementioned choice of parameters in Figure B.2 We point out that this formulation is nothing but the fully connected representation of a Convolutional layer. In the following, we will show how, the above representation opens up the perspective to a spectral interpretation of the Convolutional layer.

### B.1    Convolution parameters are eigenvalues

The first thing to point out is that convolutions are already formulated in what we could call 'reciprocal space'. Let us consider the transformation in Figure B.2. First of all, by a proper duplication of

Figure B.1: Setting the notation for dealing with CNN. In yellow the input rectangular matrix and in purple the filter structure.

Figure B.2: The figure shows how a convolution operation of Figure B.1 can be translated into a matrix-vector product via an opportune remapping of the filter into a Toeplitz matrix and a rearrangement of the rectangular image into a column vector. This is equivalent to represent the single Convolutional layer (with a single filter) as a Feed-forward fully connected one, the weight of the latter given by $\mathcal{M}(w)$.

the input components the structure of matrix $\mathcal{M}(w)$ can be simplified into one with only a single non-zero value per column.

By taking the structure of the matrix $\phi$ as the binarization version of such simplified Toeplitz matrix, the $\lambda^{in}$ of the notation introduced in [12] play the exact same role of the filters degree of freedom in the CNN layer.

We recall, indeed, that the weights strength of a feedforward fully connected layer can be written in terms of eigenvalues and eigenvectors of the equivalent directed bipartite graph as follows:

$$w_{ij} = (\lambda_j^{(in)} - \lambda_i^{(out)})\phi_{ij} \tag{B.5}$$

The effect of $\lambda^{(in)}$ can be therefore considered as a column-wise product whereas the $\lambda^{(out)}$ as a row-wise one. By setting $\lambda^{(out)} = 0$ each $\lambda^{(in)}$ can be adjusted to the same value of the convolutional weight that is solely present in a given column. This operation is graphically presented on the bottom part of Figure B.3.

## B.2 Filter relevance as eigenvalue magnitude: a bridge with existing literature

Due to equation (B.5) another correspondence can be elaborated upon: the matrix $\mathcal{M}(w)$ can be constructed using as a degree of freedom the eigenvectors components $\phi_{ij}$ as shown in Figure B.4. By setting $\lambda^{(in)} = 1$ the filters action could be accounted for by imposing an apposite structure of matrix $\phi$. The one shown in the figure is a representative example of a convolution with non overlapping filters after the input has been adjusted. The same conclusion can also be achieved for a $\mathcal{M}(w)$ structure like the one depicted in Figure B.2.

The remaining degree of freedom of the $\lambda^{(out)}$ can the be used as a measure for the filter relevance if every $\lambda_i^{(out)} = \lambda \ \forall i$.

We posit, therefore, that the the regularization proposed in [2] is equivalent to a spectral regularization for the case of a CNN with an imposed parametrization that goes as follows: every $\lambda^{(in)} = 0$, Toeplitz structure of $\phi$ and $\lambda_i^{(out)} = \lambda_j^{(out)} \ \forall i, j$.

$$\begin{pmatrix} w_1 & w_2 & w_3 & 0 & & & 0 & & w_4 & w_5 & w_6 & 0 \dots 0 & \dots w_9 \\ 0 & 0 & 0 & w_1 & w_2 & w_3 & 0 & & & & & \\ 0 & 0 & 0 & 0 & 0 & 0 & w_1 & & & & & \\ & & & & & & & & & & & \\ & & & & & \mathsf{M}(w) & & & & & & \end{pmatrix} \begin{pmatrix} x_{00} \\ x_{01} \\ x_{02} \\ x_{01} \\ x_{02} \\ x_{03} \\ \dots \end{pmatrix}$$

$$\begin{pmatrix} \lambda_1 & \lambda_2 & \lambda_3 & \lambda_1 & \lambda_2 & \lambda_3 & & \dots & \lambda_4 & \lambda_5 & \lambda_6 & & \dots & ..\lambda_9 \end{pmatrix}$$
$$\odot$$
$$\begin{pmatrix} 1 & 1 & 1 & 0 & & & 0 & & 1 & 1 & 1 & 0 \dots 0 & \dots 1 \\ 0 & 0 & 0 & 1 & 1 & 1 & 0 & & & & & \\ 0 & 0 & 0 & 0 & 0 & 0 & 1 & & & & & \\ & & & & & & & & & & & \\ & & & & & \mathsf{M}(\phi, \lambda) & & & & & & \end{pmatrix}$$

Figure B.3: In the upper part a feedforward layer equivalent to a CNN one is shown. Input components have been duplicated in order to have a single non zero component per column. In the bottom part the same weight matrix is rephrased in terms of the spectral decomposition of (B.5). The equivalence can be achieved setting $\lambda_{1,2,3...} = w_{1,2,3...}$,

$$\begin{pmatrix} w_1 & w_2 & w_3 & 0 & & & 0 & & w_4 & w_5 & w_6 & 0 \dots 0 & \dots w_9 \\ 0 & 0 & 0 & w_1 & w_2 & w_3 & 0 & & & & & \\ 0 & 0 & 0 & 0 & 0 & 0 & w_1 & & & & & \\ & & & & & & & & & & & \\ & & & & & \mathsf{M}(w) & & & & & & \end{pmatrix} \begin{pmatrix} x_{00} \\ x_{01} \\ x_{02} \\ x_{00} \\ x_{01} \\ x_{02} \\ \dots \end{pmatrix}$$

$$\begin{pmatrix} \lambda_1 \\ \lambda_2 \\ \\ \\ \lambda_{N \times M} \end{pmatrix} \odot \begin{pmatrix} \phi_1 & \phi_2 & \phi_3 & 0 & & & 0 & & \phi_1 & \phi_2 & \phi_3 & 0 \dots 0 & \dots \phi_9 \\ 0 & 0 & 0 & \phi_1 & \phi_2 & \phi_3 & 0 & & & & & \\ 0 & 0 & 0 & 0 & 0 & 0 & \phi_1 & & & & & \\ & & & & & & & & & & & \\ & & & & & & & & & & & \end{pmatrix}$$
$$\mathsf{M}(\phi, \lambda)$$

Figure B.4: The weight matrix is rephrased in terms of the spectral parameters. In this case the eigenvalues are exploited as a filter relevance proxy.

# C   Code availability

The python code used to conduct the analysis is fully available at the following GitHub repository `https://github.com/Jamba15/Spectral-regularization-teacher-student/tree/master`. The code allows both the retrieval of the full set of results presented in this work and the run of new experiments with different settings.

The implementation of the spectral layer and the Structural Pruning functions can be found in `https://github.com/Jamba15/SpectralTools`. The repository, at the moment of publication, contains a TensorFlow and PyTorch implementation with documentation.

