# OpenReview forum: "How a Student becomes a Teacher: learning and forgetting through Spectral methods"
_NeurIPS.cc/2023/Conference — NeurIPS 2023 poster_

### Official Review · Reviewer_YyTW · 2023-07-06

**Soundness:** 3 good
**Presentation:** 3 good
**Contribution:** 3 good
**Rating:** 7
**Confidence:** 4

**Summary:**

The authors propose a novel technique that allows identifying an invariant subnetwork in a student model that mirrors the characteristics of the teacher in terms of computing neurons, path distribution, and topological attributes.

**Strengths:**

 - The manuscript is clearly structured, and the subject of research is relevant
 - The authors have developed a novel technique to identify invariant characteristics of a student model mirroring key characteristics of the teacher network.

**Weaknesses:**

 - The authors have used a single synthetic dataset to perform the experiments.
 - There is little reference to related work, and no baselines are considered when comparing the proposed approach

**Questions:**

GENERAL COMMENTS:
 - (1) The authors have used a single synthetic dataset to perform the experiment. The results could be strengthened by considering multiple datasets (different synthetic parameterizations and publicly available datasets).
 - (2) We miss a more detailed related work. The authors provide a brief introduction, but references to related work are scarce. The authors may reference some works related to model distillation and pruning. In particular, we think the following works may be useful, given they mention some of the concepts and works related to those used in the manuscript:
 	- Liang, Tailin, et al. "Pruning and quantization for deep neural network acceleration: A survey." Neurocomputing 461 (2021): 370-403.
 	- (old, but relevant) Elizondo, David, and Emile Fiesler. "A survey of partially connected neural networks." International journal of neural systems 8.05n06 (1997): 535-558.
 	- Gou, Jianping, et al. "Knowledge distillation: A survey." International Journal of Computer Vision 129 (2021): 1789-1819.
 - (3) The authors should acknowledge the limitations of their research, discuss whether these limitations impact the results, and provide some insights on how these limitations could be addressed in future work.


FIGURES:
 - (4) Figure 1: enhance the wording of the caption. "Histogram for the quantities (10) (in blue) and (9) (in orange)" -> "Histogram for the quantities (Eq. 10) (in blue) and (Eq. 9) (in orange)"

SPELLING/WORDING:
 - (5) "Remarkably several topological proprieties of the teacher" -> "Remarkably several topological properties of the teacher"
 - (6) "These latter parameters are denoted λ(k) for reasons that will become clear in the following, " -> in the following?
 - (7) "The dataset will be generated by choosing the probability distribution" -> "The dataset was generated by choosing the probability distribution"?
 - (8) "values stay stably across all choices of h above 20" -> "values stay stable across all choices of h above 20"
 - (9) "To assess feature localization within the network, we generated aggregated histograms of the scalars (9) and (10)." -> "To assess feature localization within the network, we generated aggregated histograms of the scalars (Eq. 9) and (Eq. 10)."
 - (10) "To ensure that the achieved conclusions are not influenced by the size of the second hidden layer, various student configurations have been tested, yielding equivalent results." -> Please provide some details.
 - (11) "The blue histograms stands for (9)," -> "The blue histograms stands for (Eq. 9),"

**Limitations:**

The authors should acknowledge the limitations of their research and provide some insights on how these limitations could be addressed in future work.

---

> ### Author Rebuttal · Authors · 2023-08-08
>
> We would like to express our gratitude to the referee for their positive review and insightful suggestions. In relation to the 'Figures' and 'Spelling/Wording' sections, we are prepared to implement the suggested corrections should the paper progress to the camera-ready stage.
>
> Regarding the General Comments section:
>
> - _1)_ We concur with the referee on this point. We have conducted additional experiments that we are eager to incorporate into the manuscript. Specifically, in Figure 1 of the PDF, we present an analysis for various datasets: Shuffled MNIST, Shuffled Fashion MNIST, California Housing, and a more intricate Teacher structure. In the latter, the size of the first hidden layer differs from that of the second, ,removing the high degeneration of the original example and making it a more relevant case of study. The student structure remains consistent with the dimension of the second hidden layer set to 50. In this context, the performance metrics (in terms of accuracy and loss function) for both the spectral Student and the traditional student are comparable across all datasets. The findings align with the scenario described in our original paper: the spectral regularization is capable of finding a computational core (panels $a-d$) whose MSE behaviour, after perturbation, is independent from the initial layer size (panels $e-h$). Due to space limitations in the PDF, we opted against including supplementary plots, such as the loss function values and the eigenvalue histograms.
>
> - _2)_ We are in agreement with the referee on this matter. We faced challenges in locating relevant references specific to node pruning, as the majority of existing literature appears to concentrate on weight pruning. We appreciate the valuable references provided by the referee and intend to incorporate them into our manuscript in conjunction with others we have found.
>
> - _3)_ The referee's observation is on point. We will acknowledge the limitations of our study. In particular, we recognize that computational constraints prevented us from employing highly complex models where outcomes might vary. We hope to explore such scenarios in future research. Furthermore, we haven't fully explored the capabilities of the spectral decomposition. Indeed, it's feasible to conduct a feature relevance analysis in the input space. By setting $w_{ji} = (\lambda_j^{in}-\lambda_i^{out})\phi_{ji}$ in the first layer, the eigenvalues $\lambda^{in}$ could emphasize the relevance of the input's $j$-th component.
>
> We hope the referee appreciates the aforementioned changes and we thank them again for the interesting feedback.

---

> > ### Comment · Reviewer_YyTW · 2023-08-17
> >
> > We thank the authors for the rebuttal. We consider our observations have been addressed and we will keep the score.

---

### Official Review · Reviewer_dofn · 2023-07-06

**Soundness:** 4 excellent
**Presentation:** 3 good
**Contribution:** 3 good
**Rating:** 7
**Confidence:** 3

**Summary:**

This paper analyses the performance of a spectral parameterisation/regularisation scheme for neural networks. After first introducing the spectral approach, the authors describe student-teacher experiments where they attempt to distil a fixed teacher network’s behaviour into a student network. The authors show that the spectral parameterisation gives equivalently good predictive performance, but that the structure of the trained network is considerably different. In particular, they show that the spectral network shows significant sparsity when it is over-parameterised with respect to the teacher. Intriguingly, they show a way to measure the “size” of a dense computational core that seems to be invariant with respect to changing the size of the student network.

**Strengths:**

The paper is on the whole well-written and easy to understand. The authors have done a very good job of taking a theoretical topic and making it accessible and understandable.

The idea they present is simple at its core, and the results are very interesting: both that their parameterisation/regularisation scheme leads to marked sparsity and the presence of a “core” with invariant size.

While I think the results are somewhat limited (see below) the results that are presented are clearly described and care has been taken with the experimentation, leading to a convincing presentation. EDIT: the authors have fully addressed my concern around limited evaluation.

I am unable to judge the significance or novelty of these results, as I am not an expert in the field of this paper. I would say that from the perspective of a general Neurips participant that I found the results interesting and exciting. So while I would defer to experts to place the work within the literature, I will comment that I think the results may be of interest to the general Neurips audience.


**Weaknesses:**

I think the only real weakness of the paper is that the experiments are quite limited in scope. The authors use a single source of “teacher” data, which is a fully-connected network with a certain shape. I think the paper would be considerably stronger if the authors were to repeat their analysis with further sources of data. In particular, I think it would be an interesting complement to use a non-synthetic source of data as a training objective for the network, if a suitable source could be found. While this would limit the ability of the authors in making the exact correspondence between the size of the teacher and the effective size of the student, I think seeing the same behaviours - enhanced sparsity as compared to the direct parameterisation, and a “phase transition” like behaviour showing a “computational core” - would considerably strengthen the results of the paper. I think without some strengthening of the results it is difficult to accept the paper, but I think further results would change this assessment. EDIT: the authors have fully addressed my concern around limited evaluation.

While on the whole the paper displays good clarity, there are some places where the use of language leads to confusion. I have called out areas below in the questions section where I think there was particular confusion, and I think if these are addressed the paper will be sufficiently clear.

I didn’t understand the “path analysis” section, or what the significance of this result was. Perhaps being more precise with the description of the analysis would have helped (see questions).


**Questions:**

I have two non-trivial questions, and then a lot of minor comments. The two questions:

I found it surprising that the number of “active” nodes in the spectrally parameterised student networks corresponded exactly with the number of hidden nodes in the teacher network. I’m not sure how to understand this. It in some sense suggests that the data emitted from the teacher network in some sense “maxes out” the computational capacity of the network. It’s not obvious to me at all that this would happen with a randomly initialised network. In fact, it’s not really obvious what I even mean by “computational capacity” of a network. It may just be that I’m not at all an expert in these matters, but I found this confusing. Either way, I think it would be very helpful for the authors to make some comment on why such a perfect numerical correspondence might be expected, and how it is to be interpreted.

The other question, as noted above is that I didn’t understand the path analysis section. In particular I wasn’t able to understand the transformation described in line 279-281, which I think then made it difficult to understand the rest of this section. Perhaps define the transformation explicitly in terms of the components?

Some minor comments. As noted above some of these are simply English-language suggestions that I think might make the paper a little easier to read:

Line 9 (and 55): I’m not familiar with the term “quenched teacher structure” and searching the internet didn’t help me. This might be a specific technical term that I’m not familiar with, but if not I wonder whether it might be more natural to use “frozen teacher” which I think would be the more usual English idiomatic form for a teacher whose weights are not modified during the experiment.

Line 28: “self-consistent elucidation of intertwined data correlations” sounds a bit confusing, I wonder if the authors could state more plainly what they mean here?

Line 33: “spatial sorting of the chosen reservoir of computing neurons” I’m not sure what this means … what does “spatial mean” in this context?

Line 46: ​​”thus more suitable for possible hardware deployment”. This is an interesting claim. I wouldn’t have (naively) guessed that a sparse computation in eigenvalue/eigenvector space would necessarily map well to the constraints of current ML accelerator hardware. I don’t think this point needs to be expanded upon in the paper, but perhaps the authors could add a suitable reference (if one exists) that the interested reader could follow up on?

Line 75: I think I’m to read equation 1 as a simple reparameterisation. In which case it’s a little bit confusing as the w_{ij}^{(k)} here is not the same as the one introduced above. Maybe it would be clearer if the initial weight matrix were given a slightly different symbol?

Line 105: notation a bit confusing, it looks like a lower-case phi has turned into an upper-case phi. This isn’t consistent with how matrices and matrix elements are handled previously in the manuscript. It then becomes lower-case bold in equation 6 which is consistent with the previous notation.

Line 202-204: I didn’t understand what the “alternative scenario” here was. Perhaps the authors can reword this part to be more clear?

Line 215: “mean squared error (MSE) of S(h)” -> “mean squared error (MSE) of the predictions of S(h)”?

Line 226: the identification between the histogram colour and the equation here is the opposite of that in the caption of Figure 1! I think the text is the correct one.

Line 224: I’ve never heard “vehiculate” used before, and the dictionary doesn’t suggest a definition that quite fits here. Can I suggest that “drives” might be a more standard English term for this?

Figure 3 caption: describe what the colours correspond to.


**Limitations:**

I think the only considerable limitation is what I mentioned in the questions section about the weakness of evaluating with only one source of data.

---

> ### Author Rebuttal · Authors · 2023-08-08
>
> We thank the referee for the insightful feedback and in the following we will address all of their raised points.
>
> **TWO MAJOR QUESTIONS (number of active nodes and path analysis)**:
>
> In the presented results, we propose that the data originates from a 10-dimensional space and is subsequently projected in a complex manner into a 20-dimensional space, corresponding to the second and third layers. This projection can be further understood as the union of a linear function and a non-linearity (specifically, the ReLu function), particularly when biases are turned off as in our setup. Thus, we posit that at least a 20-dimensional space is necessary to capture the complete variability of the function at this stage.
>
> For clarity, the dimensions of the second student network's hidden layer is kept fixed an set at 20. Given this design (with a lot of dimension that are all equal to 20), we believe it's logical to anticipate that a well-optimized first hidden layer of the student, also of dimension 20, would be competent in identifying the salient features from the teacher's first and second embedding layers. Our rationale is further substantiated by the paper *Dynamics of stochastic gradient descent for two-layer neural networks in the teacher-student setup (Goldt et al.)*. This work, which influenced ours, points in the direction that in a teacher-student setup, the student's dimensionality should at least match that of the teacher for effective learning.
>
> In this teacher-student framework, both networks share identical activation functions and biases. This allows the linear components of the two information transfers (from the input to the first layer and from the first layer to the second) to be isolated and compared. Specifically, by examining the weights represented by $W=w^{(2)}w^{(1)}$, we can understand how activation moves linearly from the input to the second hidden layer. Each matrix entry, $W_{km}$, is defined by the equation: $W_{km}=\sum_{j=1}^{N_i=20} w^{(2)}_{kj} {w^{(1)}_j}_m$ [sorry for the misalignment but it seems there is a bug on Markdown]
>
> Here, it becomes evident that we're aggregating all the pathways from neuron $k$ to neuron $m$. Our "Path analysis" section delves into the distribution of these pathways, which represent the cumulative effects of two linear operations. The term "path" that is quite unconventional, stems from the association between linear operations and bipartite graphs.
>
> **MINOR QUESTIONS (sorted by line reference)**
> - _line 9)_ Yes, the referee is right, the term is a substitute for "frozen" and we are willing to implement the correction in the following version of the paper.
> - _line 28)_ We apologize for the lack of clarity, we are willing to rephrase this in ”organizing complex
> data by clearly understanding how different data pieces relate to each other."
> - _line 33)_ Again we are sorry for the lack of clarity. Spatial in this context is inherited from a jargon typical of dynamical systems on networks where the nodes are intended as points of a discrete space. We will modify this sentence as: "These latter reflect the spatial sorting of the chosen reservoir of computing neurons" $\rightarrow$ " These latter represent how the selected group of computing brain-like cells are organized."
> - _line 46)_ The referee is right as more conventionally the sparse implementation is the one more popular. However, there are applications where the advantage of having a smaller network structure with respect to a larger sparse one is appreciated. An example of these ML models is at the Large Hadron Collider at CERN, where the inference time required, on the order of nanoseconds, makes the smaller model preferrable. (see for example *Fast and resource-efficient Deep Neural Network on FPGA
> for the Phas EPJ Web of Conferences 245, 01021 (2020)* )
> - _line 75)_ We will change slightly the notation in the following version of the paper.
> - _line 105)_ We are sorry for the confusion, we will double check the equations and be sure that with lowercase $\phi^{(k)}$ we refer to the off-diagonal block of size $N_{k}\times N_{k-1}$ and of the square matrix $\Phi^{(k)}$ of size $(N_{k-1} + N_k)\times (N_{k-1} + N_k)$.
> - _line 202-204)_ The alternative scenario we are referring to, which will be also included is something we are willing to analyze in follow-up papers and it is the effect of adding also the input eigenvalues, labelled $\lambda^{(in)}$ as trainable variables. This will lead (for the input layer) also to the possibility of finding the relevant input space features by inspecting their after-training modulus (of course employing the regularization).
> - _line 215)_ We agree that it would be better rephrased in the suggested way.
> - _line 224)_ We can rephrase the word in the suggested way.
> line 226) Again, the referee is right, indeed there is a mistake. The Blue one, that gets more peaked in zero is the one of the spectral layer, eq. (10).
> - _Figure 3)_ We are willing to implement the suggested modification in the following version.
>
> Moreover, we have extended our analysis to 4 more relevant and complex datasets, more specifically in Figure 1 of the PDF page we show the same analysis carried out with for: Shuffled MNIST, Shuffled Fashion MNIST, California Housing and a more elaborated Teacher structure. In the latter, the size of the first hidden layer is different from the one of the second hidden. The student structure is left unchanged and the performances (in terms of accuracy and loss function) of both, the spectral Student and the classical student, are the same in each dataset. The results shown are in line with the simplified scenario that we have presented in the original version of the paper and we believe extend considerably the validity of our results.
>
> We hope that, in light of the aforementioned changes, the referee will reconsider the rating given and we thank them again for the feedback.

---

> > ### Comment · Reviewer_dofn · 2023-08-14
> >
> > I thank the authors for responding in detail to my comments and for taking the feedback into account when revising the manuscript.
> >
> > I think the main weakness that I had identified - the use of a single, synthetic dataset - has been convincingly overcome by the authors' addition of several real-world dataset. I am pleased to see that the same qualitative "computational core" behaviour holds for these datasets, and I think the results section now appears very strong.
> >
> > It strikes me that as, an additional optional change, the authors could perhaps show on the right hand side of their revised figure 1 (panels e-g) what happens if units are randomly removed from the "direct" network. This is first of all just to confirm that non-spectral neural networks don't show similar effects on (random) node deletion (very unlikely, I don't see how they possibly could), but also would show a stark contrast with the authors' method, further highlighting its strength.
> >
> > Overall I think the addition of further results has considerably strengthened the paper, and I am happy to change my recommendation.

---

### Official Review · Reviewer_Cz6u · 2023-07-06

**Soundness:** 3 good
**Presentation:** 4 excellent
**Contribution:** 3 good
**Rating:** 6
**Confidence:** 2

**Summary:**

The authors consider a knowledge distillation setting involving a teacher and a student neural network.
The authors exploit a few interesting tricks -especially the use of a spectral parameterization of the network, and special regularizers- to show that it is possible to enforce learning a submodule within a student network (as long as it is larger than the original teacher), that implements the teacher behavior with a minimal number of neurons, and which can thus be used to estimate the effective size of the teacher.

The authors present thorough theoretical work, along with experimental verification; the results are convincing, but the topic is outside my area of expertise.

**Strengths:**

- The paper is interesting and well written.
- The results are convincing.


**Weaknesses:**

- The empirical experiments could be more extensive.
- The dataset used is extremely simple, and it is not clear whether the method would work in more typical scenarios.


**Questions:**


N/A


**Limitations:**

- Evaluation is performed only on small networks and on toy data. It remains to be seen how well the approach would work on complex, modern datasets and network architectures.

---

> ### Author Rebuttal · Authors · 2023-08-08
>
> We express our gratitude to the referee for their feedback. We acknowledge the need for more comprehensive results. In response, we've conducted four additional more complex scenarios, with the outcomes presented in Figure 1 of the accompanying PDF file. The experimental framework remains consistent with the original paper, and the outcomes corroborate the preliminary findings we described. In more detail, our expanded analysis encompasses four unique datasets: Shuffled MNIST, Shuffled Fashion MNIST, California Housing, and a nuanced Teacher structure. In this Teacher model, the first hidden layer's size is distinct from the second and from the fixed one of the student. We've adhered to a uniform student structure, with the only variation being the second hidden layer's size set to 50. At variance with the showcase of the paper, however, we are not able to set a proper complexity threshold in the plots. The performance indicators, both in accuracy and loss function, for the spectral Student and the classical Student, remain consistent across all datasets.
> We would like to stress that with those datasets that are much closer to the typical scenarios in terms of complexity and input dimension, the capability of finding the computational core of the student (panels $a-d$) and the phase transition-like behavior of the MSE (panels $e-h$) are both validated.
>
> We trust that in light of these additional findings, which in our view enhance the manuscript's robustness, the referee might reconsider the manuscript's rating.
>
> Regarding the network architecture, we intend to address this in the _Limitations_ section of the paper and are keen to delve into more sophisticated architectures, such as Residual Networks and Transformers, in forthcoming publications.

---

> > ### Comment · Area_Chair_t9d6 · 2023-08-18
> > **Thanks for authors' rebuttal!**
> >
> > Reviewer Cz6u, did the authors address your concerns on simple dataset and experiments? Thanks.

---

### Official Review · Reviewer_EH9K · 2023-07-06

**Soundness:** 2 fair
**Presentation:** 2 fair
**Contribution:** 2 fair
**Rating:** 6
**Confidence:** 1

**Summary:**

This work focuses on the teacher-student paradigm in theoretical machine learning and shows that, for a unique optimization scheme that involves directly optimizing on the eigenvalues/eigenvectors of the data, a stable subnetwork in the student can be identified that can mirror the complexity of the teacher network. The area of this work is outside of my domain, so I am unable to comment further on the contributions.

**Strengths:**

The approach seems theoretically-motivated and the result seems interesting. The area of this work is outside of my domain, so I am unable to comment further on the contributions.

**Weaknesses:**

The experiments seem limited, but the area of this work is outside of my domain, so I am unable to discern what level of experimentation is normal for this kind of work.

**Questions:**

How can the spectral method be scaled up for large datasets, like images?

**Limitations:**

I did not see a section dedicated to limitations of the authors' work.

---

> ### Author Rebuttal · Authors · 2023-08-08
>
> We appreciate the referee's feedback and recognize that the work may not fall directly within their domain of expertise. Regarding the implementation with other data, we agree with the referee and therefore have extended our analysis to four more complex and realistic datasets. Delving deeper, our refined analysis covers: Shuffled MNIST, Shuffled Fashion MNIST, California Housing, and an intricate Teacher structure. Within this Teacher design, the size of the first hidden layer differs from that of the second. We've maintained a consistent student framework, the only deviation being a second hidden layer size of 50. For both the spectral Student and the conventional student, performance metrics, encompassing accuracy and loss function, are uniform across all datasets.
>
> Regarding the implementation of a large image dataset in the main paper we show how the spectral parametrization can be extended to a Convolutional Layer. In this setting, the eigenvalues can be mapped into weights that leverage the relevance of every filter resulting in an already analyzed algorithm of proven effectiveness.
>
> We are, moreover, willing to insert a dedicated Limitation section where we recognize that due to computational limitations, we haven't tested more complex models where outcomes might differ. Nevertheless, we aspire to explore such models in forthcoming research. Furthermore, we haven't fully tapped into the capabilities of spectral decomposition. Indeed, a feature relevance analysis can also be conducted in the input space.  By setting $w_{ij}=(\lambda_j^{in}-\lambda_i^{out})\phi_{ij}$ in the initial layer, the eigenvalues $ \lambda^{in} $ could highlight the significance of the input's $j$-th component.
>
> We hope that, given these supplemental results which we believe bolster the paper's strength, the referee might re-evaluate the manuscript's current rating which we believe is a little unfair considering their assessment of inexperience with the domain.

---

> > ### Comment · Reviewer_EH9K · 2023-08-14
> >
> > Thanks for adding the new experimental results. I believe this strengthens the work and I have raised my score (though I encourage the AC to give the other reviews more weight, since I have kept my confidence score as a 1).

---

### Official Review · Reviewer_uE1J · 2023-07-22

**Soundness:** 2 fair
**Presentation:** 2 fair
**Contribution:** 2 fair
**Rating:** 5
**Confidence:** 3

**Summary:**

This paper tried to understand and extend a new parametrization, spectral parametrization, for fully connected networks. They empirically show that in the teacher student setup, even when the student network is highly over parametrized, the student network under that parametrization will converge to a somehow "sparse" network that can be compressed, using standard optimizers. And they show that standard parametrization for the student network cannot do that.

**Strengths:**

No.

**Weaknesses:**

1. Lack novelty. Why this parametrization can lead to sparsity is almost well-understood in the literature. For instance, it's well known that for such model $h_{\theta}(x)=h(x;u\odot v)$, if we initialize $u,v$ to be small and use standard gradient based algorithms, we will have a gradual rank increase for $u,v$, which leads to sparsity more easily. For some reference we can look at Abbe's paper https://arxiv.org/abs/2306.07042 (This is the most recent reference) or Jason's paper https://arxiv.org/abs/2207.04036. Or we can simply compute GD dynamics for using diagonal linear network to learn a linear target with small initialization. In a word, this result is not surprising to me. I feel like I already know this/ expect that happens.

2.For the empirical result, the input dimension in the experiments is too low. It's only 10. We have high dimensional input in practice. Also, the experiments are insufficient. Like you can try different optimizers, try different hyperparameters (width, initialization scheme), etc.

3.The authors didn't discuss the related works sufficiently. This phenomenon is clearly related to training dynamics of fully connected networks and there are many theory papers discussing the same things (even similar implication/conclusion).

Update: since the authors update the experiments, I have changed my score.

**Questions:**

No.

**Limitations:**

I don't think they discuss their limitations adequately. Here are my suggestions for improvement.

Since this phenomenon on simple network structures is actually almost well understood, if you really want to show this kind of reparametrization works, you should do some larger scale experiments. If there are some larger scale experiments to show that that kind of parameterization really works in practice and helps practice problems, then I think it's a good paper.

---

> ### Author Rebuttal · Authors · 2023-08-08
>
> We would like to thank the referee for pointing us to the insightful reference concerning implicit regularization resulting from the commuting nature of our parametrization. Unfortunately, the other mentioned reference came out after the submission of our work to the conference. However, we contend that the effect we describe extends beyond such implicit regularization. Specifically, we do not initialize our spectral weights to small values; instead, they are uniformly set to one in every trial. This suggests that rather than imposing a gradual increase in rank, we observe more of a decrease. While our parametrization may indeed exhibit an implicit bias towards sparse representation, it alone cannot account for the observed distribution of $\\lambda_i$ and consequently, $\\mathcal{L}_i$.
>
> In the provided PDF, we present the distribution of $\\mathcal{L}_i$ (very similar to the one of $\\lambda_i^{(out)}$) for Shuffled Fashion MNIST. Notably, the after training distribution of the variable with explicit $L_2$ regularization is very different from the one we get whithout, relying solely on the implicit bias. The visual inspection of the image points in the direction that is the eigenvalues' sparsification, leading to feature-centric sparsification (involving node elimination rather than link elimination), which is considerably more pronounced when explicit bias is applied.
>
> Regarding the dataset's simplicity, we concur with the referee's observation and have undertaken additional experiments that we are willing to include in the manuscript. Specifically, in Figure 1 of the PDF, we showcase our analysis on four distinct datasets: Shuffled MNIST, Shuffled Fashion MNIST, California Housing, and a more intricate Teacher structure. For this Teacher configuration, the size of the first hidden layer differs from the second. We maintained a consistent student structure, and performance metrics (in terms of accuracy and loss function) for both the spectral Student and the conventional student are consistent across all datasets. We point outh that, in this case, we are not able to set an obvious complexity threshold of the teacher.
>
> The results presented align with the simplified scenario depicted in the original version of our paper and we hope that, in light of those new findings, the referee will be keen on reconsidering the given 'Rating'.

---

> > ### Comment · Reviewer_uE1J · 2023-08-14
> > **Require Further Explanations**
> >
> > I appreciate you providing this feedback. Upon further consideration, I now understand that your initialization is outside of the small initialization regime, which is nice.
> >
> > My primary concern at this point is regarding the scale of the dataset utilized. Have you previously experimented with substantially larger datasets such as CIFAR-100 or ImageNet? If your methodology achieves similar performance on those larger datasets, I would significantly increase my assessment score to at least 5.

---

> > > ### Author Response · Authors · 2023-08-16
> > > **Results with larger datasets**
> > >
> > > We are pleased to hear that the reviewer appreciates our results. We acknowledge the concerns raised about the behavior with larger datasets. To address this, we have evaluated our method on CIFAR-100, employing a pretrained ResNet50 backbone enhanced with three fully connected Spectral layers and Batch Normalization. Even in this more complex setting all the results concerning the presence of an invariant core of information are in line with the ones of simpler scenarios, presented in the provided 'pdf'.
> > > We are prepared to incorporate these supplementary results in our paper's subsequent version and hope that, with these clarifications, the reviewer might positively reconsider their evaluation of our work.

---

> > > > ### Comment · Reviewer_uE1J · 2023-08-19
> > > > **Thanks**
> > > >
> > > > That's great. I will increase my score to 5.

---

### Author Rebuttal · Authors · 2023-08-08

We express our gratitude to the chairs and referees for their valuable effort in providing constructive feedback on our manuscript. We have carefully considered each of the comments made and addressed them individually and in-depth in the replies to the referees. We have taken note that all the referees highlighted the need for more than one teacher and the use of more complex and realistic datasets. We acknowledge that this could be a weakness of our work. To address these concerns and enhance the quality of our paper, we conducted various new experiments on different datasets and with a more elaborate teacher structure. The results of this supplementary analysis are available in the attached PDF and demonstrate that all the spectral regularization effects still persist in these complex scenarios.

For this scope, we have analyzed various datasets, including Shuffled MNIST, Shuffled Fashion MNIST, California Housing, and a more complex Teacher structure. Both the spectral Student and classical one achieved similar accuracy and losses across all datasets. Although we regret not being able to include additional plots, such as the loss function value and eigenvalue histogram, due to space constraints in the PDF, the plots show the clear presence of a regularization effect in the spectral parametrization irrespectively of the initial size of the Student and of the task assigned to it (see panels $a-d$). Moreover the computational core found behaves in the same way when its neurons are removed (panels $e-h)$

These additional findings will surely strengthen the claims of our paper and will be incorporated into the main manuscript if it is accepted. Hoping that these new results and the point-by-point reply that we have given to each referee can convince them of the quality and robustness of our work, we remain.

Respectfully yours,
The authors

---

### Decision · Program_Chairs · 2023-09-21

**Decision:**

Accept (poster)

**Comment:**

Reviewers agree that the paper addresses an interesting question regarding teacher-student setting. and demonstrates results on both synthetic and real-world dataset (after rebuttal). Reviewers are satisfied with the rebuttal. I would encourage the authors to further improve the presentation, in particular by writing down formal definition and algorithms, and properly incorporating new results in the camera ready.